# Comparing the effectiveness of computer-aided design/computer-aided manufacturing (CAD/CAM) of insoles manufactured from foam box cast versus direct scans on patient-reported outcome measures: a protocol for a double-blinded, randomised controlled trial

Laura Barr,[1,2] Jim Richards [iD],[2] Graham J Chapman [iD][2]

¹Orthotic Department, Gartnavel General Hospital, Glasgow, UK
²Allied Health Research unit, University of Central Lancashire, Preston, UK

**Correspondence to**
Dr Graham J Chapman;
GChapman2@uclan.ac.uk

## ABSTRACT

**Introduction** Custom insoles are a routine treatment for many foot pathologies, and the use of computer-aided design and computer-aided manufacturing (CAD/CAM) is well established within clinical practice in the UK. The method of foot shape capture used to produce insoles varies throughout orthotic services. This trial aims to investigate the effectiveness of two common shape-capture techniques on patient-reported outcomes in people who require insoles for a foot or ankle pathology.

**Methods and analysis** This double-blinded randomised controlled trial will involve two intervention groups recruited from a National Health Service orthotic service. Participants will be randomly assigned to receive a pair of custom CAD/CAM insoles, manufactured either from a direct digital scan or a foam box cast of their feet and asked to wear the insoles for 12 weeks. The primary outcome measure will be the Foot Health Status Questionnaire (FHSQ) pain subdomain, recorded at baseline (immediately after receiving the intervention), 4, 8 and 12 weeks post intervention. Secondary outcome measures will include FHSQ foot function and foot health subdomains recorded at baseline, 4, 8 and 12 weeks. The Orthotic and Prosthetic User Survey Satisfaction with Device will be recorded at 12 weeks. The transit times associated with each arm will be measured as the number of days for each insole to be delivered after foot shape capture. Tertiary outcome measures will include participant recruitment and dropout rates, and intervention adherence measured as the daily usage of the insoles over 12 weeks. The change in FHSQ scores for the subdomains and insole usage will be compared between the groups and time points, and between group differences in time in transit, cost-time analysis and environmental impact will be compared.

**Ethics and dissemination** Ethical approval was obtained from the Health Research Authority, London Stanmore Research Ethics Committee (22/LO/0579). Study findings will be submitted for publication in peer-reviewed journals, conference presentations and webinars.

| STRENGTHS AND LIMITATIONS OF THIS STUDY |
| :--- |
| ⇒ This trial aims to investigate routine interventions that are used within the National Health Service, therefore the outcomes have the potential to impact on the decision-making within orthotic services across the UK and beyond. |
| ⇒ The double-blinded design reduces the risk of bias from the participant and the investigator. |
| ⇒ Single-centre design may limit the cohort. |
| ⇒ The multiple follow-up points may lead to drop outs. |

**Trial registration number** NCT05444192.

## INTRODUCTION

The integration of computer-aided design (CAD) and computer-aided manufacturing (CAM) technologies has increased in prevalence in the orthotic industry over the past two decades.[1] The manufacture of bespoke foot orthoses, commonly known as insoles, involves capturing an image of the foot from which an insole is ultimately created. A fully digital system involves replacing each step of the traditional manufacture process with a computer-aided counterpart. As such, the manufacture of insoles using fully digital systems involves taking a digital scan of the foot to create a three-dimensional (3D) model, as opposed to the traditional method which commonly involves capturing a physical impression of the foot using a foam box or plaster cast.[2] The evolution of insole production with the introduction of CAD/CAM has led to the creation of two industry standards within National Health Service (NHS) orthotic services across the UK with

some services manufacturing CAD/CAM insoles using traditional foam box casts, while others have chosen to use direct digital scanning.

Digitisation of orthotic devices conceptualised gains in production speed and reduction in waste materials when compared with traditional manufacture using physical shape capture.[3] Yet, the continued interim stages of foam box casting and physical transportation of these foam box casts to manufacturers, which is required by services who do not own or have access to scanning equipment, sacrifice these benefits. Motivation and hesitation in transitioning to a fully digital workstream have been assumed, but are currently unsupported in the literature. A common reluctance to adopt direct digital scanning is based on the assumption that direct digital shape capture and foam box casts do not produce like-for-like models, and therefore cannot result in the production of equally effective insoles. Although differences in volume have been shown,[4–6] differences in the accuracy and reliability between methods remain unclear,[7] and ultimately differences in the treatment efficacy and resulting effectiveness have not been evaluated. Other concerns centralise on costs associated with the acquisition of direct digital scanning equipment. Although prior cost analyses have shown a fully digital supply chain to be more expensive than a fully traditional supply chain,[3] this does not reflect the practices associated with a partially digital workflow as seen across the NHS. Furthermore, it does not consider a cost comparison over the lifespan of a direct digital scanner, or the environmental impact of the manufacture, transportation and disposal of traditional cast materials.

Overall, the evidence base relating to CAD/CAM insoles demonstrates little consistency or rationale behind the mode of shape capture used during the insole manufacturing process. Often the shape-capture method is undocumented or unclear,[8] or documented without any attributed clinical reasoning.[9–11] In 2019, Parker *et al*[3] investigated the differences in a fully digital workflow compared with fully traditional manufacturing techniques, but did not investigate the specific impact of shape capture in isolation. Furthermore, the aforementioned studies report no consideration as to the environmental impact of phenolic foam production and disposal required for traditional foot shape capture,[12 13] or the carbon footprint of transportation from manufacturer to digital upload of the foot shape from the foam box cast, a step which is not required when using direct digital scan techniques. In line with NHS Net Zero targets,[14] and the recognition of orthotic services throughout the UK that a large-scale change is required to achieve this,[15] the practice of single-use traditional shape-capture techniques requires scrutiny. It is clear from the literature and current widespread indiscriminate practices across NHS orthotic services that more research is required to assist with best practice and decision-making in the manufacture of CAD/CAM insoles.

In NHS Greater Glasgow and Clyde (GGC), the largest NHS Health Board in Scotland, CAD/CAM insoles represent 14% (n=2739, per annum in 2020) of all orthotic department provision. Assuming a similar proportion throughout the rest of the UK, this represents a significant proportion of orthotic service users and financial burden to the NHS. Given the proportion of orthotic service users who receive insoles from NHS orthotic services, this trial has the potential to guide practice towards beneficial changes in patient outcomes, as well as providing NHS orthotic departments with information to assist in the development of long-term service models in line with NHS and Government efficiency and net zero targets.

## Study aims and objectives

The aim of this trial is to evaluate the clinical effectiveness of two commonly used foot shape-capture techniques for the manufacturing of custom CAD/CAM insoles over 12 weeks. The primary objective is to compare the changes in self-reported foot pain between two groups of participants randomly assigned to receive custom CAD/CAM insoles manufactured from direct digital foot scan or foam box casting. We hypothesise that the outcomes will be equivalent for the two methods. Secondary objectives include comparing the changes in foot function, foot health, satisfaction with treatment, time in transit, cost-time analysis and environmental impact between the two methods.

## Trial design

We propose a single-centre, double-blinded, randomised controlled trial comparing the effectiveness of two methods of foot shape capture to manufacture custom-made insoles. This is an interventional, equivalence trial using medical devices commonly known as insoles.

## METHODS AND ANALYSIS
## Study setting

Participant assessment and treatment will be provided in a hospital setting, within the NHS GGC orthotic department. There will be one trial site located at the Glasgow Royal Infirmary. This trial will minimise participant on-site visits by using telephone contacts throughout the participation period to collect relevant participant reported outcome measures. At the conclusion of the participants' involvement in the trial, they will transition back to usual care within the NHS GGC orthotic department. The study protocol has been reported in accordance with the Standard Protocol Items: Recommendations for Interventional Trials guidelines.[16]

## Participant recruitment

Individuals referred to the NHS GGC orthotics service with a musculoskeletal (MSK) medical condition or lower limb biomechanical deficit which would commonly be treated with the use of insoles as a first or second line intervention following the NHS GGC MSK Foot and Ankle Pathway[17] will be offered the opportunity to enrol

**Table 1** Participant eligibility criteria

| Inclusion | Exclusion |
|---|---|
| ▶ Aged 18 years or above.<br>▶ Referred to the NHS GGC orthotic service requiring a new assessment for insoles.<br>▶ Deemed suitable for CAD/CAM insoles as assessed by the PI or Co-I on clinical assessment.<br>▶ Able to commit to five appointments over a 16-week period (two face-to-face appointments and three telephone appointments).<br>▶ Have suitable own outdoor footwear that can accommodate a CAD/CAM insole as assessed by the PI or Co-I and can wear these for 12 weeks in accordance with standard practice.<br>▶ An adequate understanding of written and verbal information in English in order to provide informed consent and answer the study questionnaires. | ▶ Scheduled elective surgery or other procedures which is likely to affect mobility during the trial.<br>▶ Scheduled steroid injections to the foot or ankle up to 3 months prior to joining or during the trial.<br>▶ Aged<18 years.<br>▶ Adults with incapacity, under The Adults with Incapacity (Scotland) Act.<br>▶ Participant unable or unwilling to consent.<br>▶ Medial longitudinal arch height of the foot exceeds depth of EVA blank (35 mm).<br>▶ Clinical assessment concludes that the participant requires an insole material other than EVA.<br>▶ Clinical assessment concludes that the participant does not require or will be unlikely to benefit from CAD/CAM insoles, as outlined in the NHS GGC Foot and Ankle Pathway.[17]<br>▶ The participant is unable to commit to the trial conditions.<br>▶ Peripheral neuropathy present.<br>▶ Active foot ulceration present.<br>▶ Participants with life expectancy of less than 6 months.<br>▶ Any other significant disease or disorder which, in the opinion of the PI or Co-I, may either put the participants at risk because of participation in the trial, or may influence the result of the trial, or the participant's ability to participate in the trial.<br>▶ Participants who have participated in another research trial involving an investigation of foot orthosis in the past 12 weeks. |

CAD, computer-aided design; CAM, computer-aided manufacturing; Co-I, co-investigator; EVA, ethylene vinyl acetate; GGC, Greater Glasgow and Clyde; NHS, National Health Service; PI, primary investigator.

in the trial. In order to provide a realistic representation of day-to-day clinical practice, participants' pathology will not be limited to one specific pathology, a similar approach taken in other studies investigating orthoses for non-specific lower limb MSK pathologies.[10 18 19] Table 1 provides detail of the inclusion and exclusion criteria.

### Assessments

Figure 1 shows the study flow chart for eligible participants. At the initial visit, participants will attend a face-to-face hospital visit to be assessed and screened according to the inclusion and exclusion criteria. Having read the participant information sheet for the trial and having the opportunity to ask further questions, participants will sign a consent form and formally enrol in the trial. During the baseline assessment, relevant medical history will be recorded, as well as any routine medications taken by the participant. Physical examination of the foot and ankle will include the Foot Posture Index-6,[20] Jacks test for functional hallux limitus,[21] palpation technique for subtalar joint axis location,[22] passive assessment of ankle dorsiflexion stiffness by position of first detectable resistance,[23] supination resistance test[24] and a visual gait analysis in the sagittal and coronal planes. Following the clinical assessment, participants will undergo both a direct digital scan and foam box cast of their feet so

that the participants are unaware of which manufactured insole group they will be randomly assigned to. Direct digital scans will be acquired using the Paromed ParoScan 3Dm mobile 3D scanner and foam box casts will be taken using 6 cm deep 'Foot Impression Boxes' (Algeos, UK). In order to minimise any differences between casting and scanning methods, all foam box casts and direct digital scans for all participants will be taken by the primary investigator (PI) who has over 15 years' experience in the assessment, shape capture and design of insoles. All foam box casts and direct digital scans will be taken in a semiweight bearing position, with the participant seated, and shape capture undertaken one foot at a time, with the contralateral foot positioned on the floor. The foot will be manipulated by the clinician into the optimal position as determined by the participant's clinical assessment, the presenting MSK pathology and the Foot Posture Index (FPI), before being placed into the foam box and the scanner. For example, in instances of pathologies affecting the medial aspect of the foot, ankle or leg, and where FPI values are between 0 and +12, an external rotational force will be applied to the participants leg by the clinician, effectively supinating the foot in the foam box cast and on the direct digital scanner. Conversely, in instances of pathology affecting the lateral

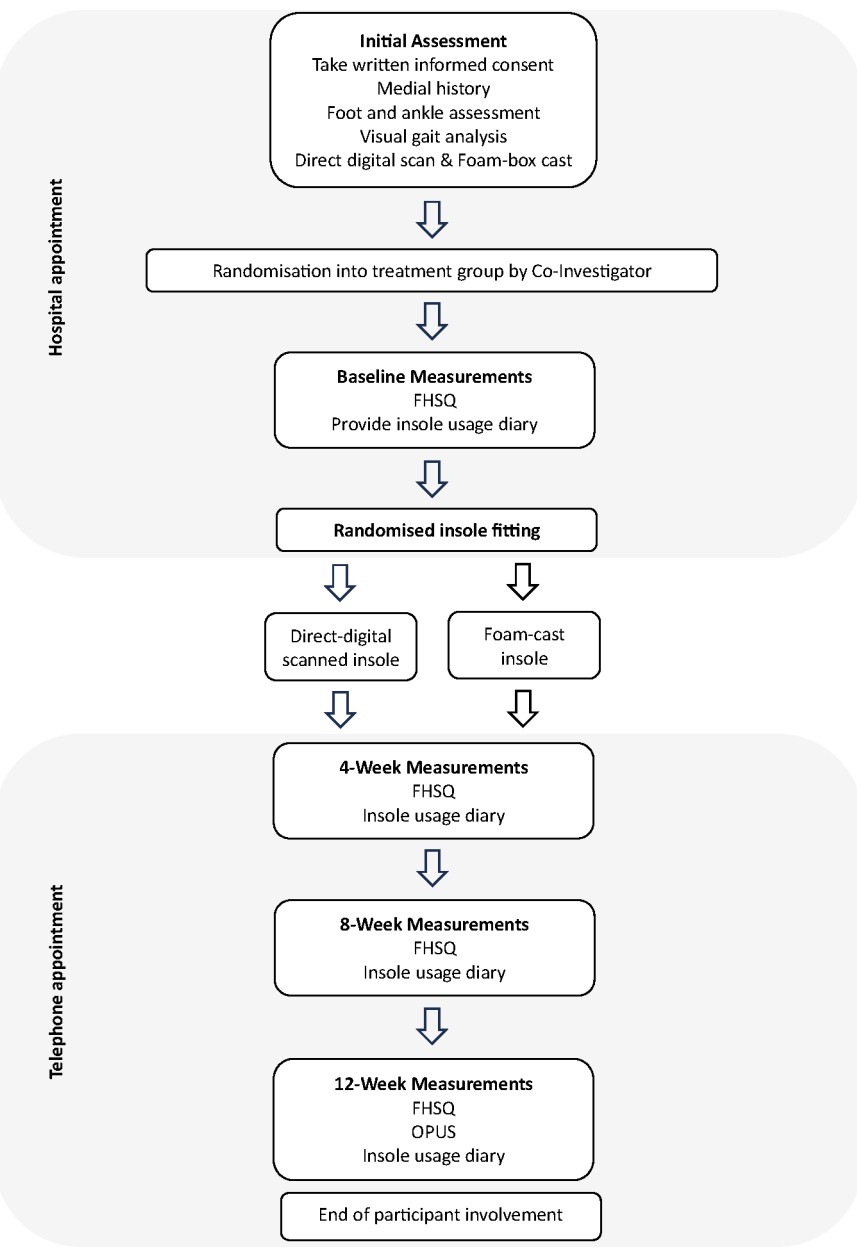

**Figure 1** Study flow chart for eligible participants. FHSQ, Foot Health Status Questionnaire; OPUS, Orthotic and Prosthetic User Survey Satisfaction with Device.

aspect of the foot, ankle or leg, and where FPI values are between 0 and −12, an internal rotational force will be applied to the participants leg by the clinician, effectively pronating the foot in the foam box cast and on the direct digital scanner. Where the participant has mobility of the first ray and the insole prescription is to be designed to facilitate first ray plantarflexion, the clinician will manipulate the first ray into a plantarflexed position by applying a downward force to the first metatarsal head in the foam box cast and onto the direct digital scanner. These example techniques described for positioning the leg, foot and first ray are similar to those described in previous literature regarding the effectiveness and repeatability of casting and scanning techniques.[10 25]

## Randomisation

At the end of the initial visit, participants will be randomised to either the direct digital scan or foam box cast manufactured insole group. Randomisation will be conducted according to a random number algorithm, contained in presealed envelopes. The envelopes will be opened on a 1:1 basis by the co-investigator (Co-I). The PI and the participants will be blinded to the treatment arm.

In the event of a participant experiencing an adverse event and/or the medical condition of a participant necessitates unblinding, a Co-I (not blinded to the randomised intervention) will access the CAD/CAM insole ordering system to confirm the treatment arm. This process will

not unblind the whole trial, nor will it disclose the randomisation schedule.

## Interventions

All participants will receive a pair of custom CAD/CAM ethylene vinyl acetate (EVA) insoles. The insoles will be manufactured from the allocated randomised technique; either direct digital scan or foam box cast which will then be scanned into the CAD/CAM system (ParoManager Paro360 V.1.99, Paromed, Germany). All scanned CAD/CAM images will then be modelled by the PI, who is blinded to the treatment arm, and has over 15 years' experience using the ParoManager CAD/CAM modelling system. The insole prescription and design will be conferred by the PI and the Co-I who assessed the participant. The authors acknowledge that it is not possible to design a prescription protocol due to the variety of presentations that will be recruited to the trial. In accordance with standard practice, insole prescription will be determined by the physical and biomechanical assessment for each participant and will be conferred by two experienced clinicians at the time of assessment. Prescriptions are likely to include a variety of functional design features, for example, this may include the use of medial heel wedging for participants presenting with medial foot, ankle or lower limb pathology,[26 27] and medial heel skives may be considered if participants do not present with plantar heel pain.[28] Medial forefoot wedges will be considered for participants presenting with medial foot or ankle pathology and a concurrent forefoot varus.[29] Conversely, the use of lateral forefoot wedges may be included for participants with a lateral foot or ankle pathology.[30] Heel raises will be considered where there is reduced range of ankle dorsiflexion, posterior or plantar heel pain, or leg length discrepancy.[31] Metatarsal domes may be considered in conjunction with other functional design elements for participants with plantar forefoot pathology.[32] The EVA Shore hardness will be determined by the individual characteristics of the participant assessment. Participants with moderate-to-high supination resistance score or medially deviated subtalar joint axes will be considered for harder EVA insoles (50–70 Shore). Those participants with a low supination resistance score will be considered for insoles with a hardness 30–40 Shore. Participants with characteristics such as forefoot plantar fat pad atrophy will be considered for mixed 30/50 or 50/70 Shore EVA, where the Shore harness at the forefoot is softer.

## Follow-up

After 3 weeks of the initial assessment, participants will return to the hospital setting for their insoles to be fitted. At this appointment, baseline outcome measures will be collected and participants will be provided with a diary to record their daily insole use. Outcome measures and insole use will be collected via telephone appointments at 4, 8 and 12 weeks post baseline. Any issues with the insoles can also be raised by participants at these time points, and appropriate action will be taken by the research team to resolve and record any issues or adverse events that may have arisen. Participants will also be provided with contact details for the NHS GGC orthotic department and the research team, to raise any issues out with these time points, and appropriate resolution will be agreed and recorded on a case by case basis.

## Outcome measures

Patient-reported outcome measures will be collected at baseline, 4, 8 and 12 weeks of insole use. The Foot Health Status Questionnaire (FHSQ) has validated subdomains for pain, function and foot health,[33–35] which will be completed at each time point. These will be considered using inferential statistics and minimal important differences.[36] The primary outcome measure will be the FHSQ pain subdomain, with the subdomains for function and foot health being used as secondary outcome measures. The Orthotic and Prosthetic User Survey Satisfaction with Device (OPUS) will also be used as a secondary outcome measure, completed after 12 weeks of insole use, to evaluate the patient satisfaction.[37 38] A further secondary outcome measure will include the time in transit, measuring the number of days for each insole to be delivered to the trial site after foot shape capture. This will allow an analysis of insole production in NHS GGC during the trial period, relating to the environmental impact of the required phenolic foam production[13] and carbon footprint of transportation from manufacturer to digital upload of foam box cast using carbon footprint calculations.[12] Tertiary outcomes will include measurement of the recruitment rate and dropout rate for the duration of the trial, and participant adherence to the trial protocol whereby participants will be asked to keep a diary of daily wear time, in accordance with prior publications on measuring orthotic adherence.[39] The minimum threshold for adherence for this trial is considered to be >21 hours per week.[40]

## Sample size

A sample size power calculation, based on data from Landorf et al[36] regarding FHSQ, was used to detect a clinically important difference between groups of 13 (SD=26.9) points in FHSQ scores using the pain subdomain as the primary outcome. Giving a required minimum sample size of 54 participants in each group and including a 5% dropout rate, 57 participants will be recruited into each group, thus requiring a total sample size of n=114.

## Data management and auditing

On entering the study, participants will be given a unique trial number to ensure participant anonymity throughout the trial. The unique trial numbers and participant details will be stored securely and separate from the project files. All data and personal information will comply with the requirements of the General Data Protection Regulation 2018. Data handling will comply with standard operating procedures of the trial sponsor (NHS GGC) and the

University of Central Lancashire. Trial monitoring will be conducted by the sponsor (NHS GGC).

## Adverse events

This trial is considered a low-risk trial for adverse events by the sponsor (NHS GGC). Any adverse events will be recorded and reported to the trial sponsor.

## Withdrawal of participants from study

During the course of the trial, a participant may choose to withdraw from the trial at any time. This may happen for a number of reasons, including but not limited to the occurrence of what the participant perceives as an intolerable adverse event, inability to comply with trial procedures or participant decision without reason. In addition, the PI may discontinue a participant from the trial treatment at any time if the PI considers it necessary for any reason including, but not limited to ineligibility arising during the trial that is, development of a medical condition as outlined in the exclusion criteria, or significant non-compliance with treatment regimen or trial requirements that is, participant has not worn or unable to wear the insoles between appointments. The type of withdrawal and reason for withdrawal will be recorded in the case report form.

## Missing data

Missing follow-up data for the primary and secondary outcome measures are likely to be minimal, with missing data potentially due to participant dropout. If one or more observations are missing, the last observation recorded will be carried forward in the primary analysis; however, for those patients who dropout, at least one follow-up time point will be required for the data to be carried forward. Data for participants who do not reach the minimum self-reported adherence threshold of >21 hours per week, calculated as an average across the 4-week, 8-week and 12-week time points, will still be included in the final between-group analysis to establish if adherence differs between groups.

## Patient and public involvement

The study protocol and documentation were prepared with input from five patients who attended the NHS GGC orthotics service. On reviewing the patients' constructive feedback, the study design was refined to incorporate telephone follow-up appointments to minimise participant commitment to face-to-face appointments.

## Statistical analysis

The primary and secondary outcome measures will be compared between groups at the specified data collection time points: baseline, 4, 8 and 12 weeks, using mix methods analysis of variance or Friedman tests with post hoc Wilcoxon tests for within group analysis and Mann-Whitney U tests for between group analysis if the data are not normally distributed. The change in OPUS scores for the subdomain of satisfaction with device will be compared between groups at the specified data collection time point of 12 weeks, using unpaired t-tests or Mann-Whitney U tests depending on the data distribution.

## Ethics and dissemination

Ethical approval has been obtained from London Stanmore Research Ethics Committee (22/LO/0579), and the trial is registered on ClinicalTrials.gov and any protocol amendments will be numbered and uploaded to this site. This trial has been written and will be performed according to the Declaration of Helsinki. The results from the trial will be presented at national and international conferences, webinars and published in peer-reviewed journals. Authorship eligibility will be based on the recommendations from the International Committee of Medical Journal Editors.

## Data sharing

Data generated from this study will be made available for research and academic purposes, after the publication of the trial results, on request via email to the corresponding author. Available data will include anonymised individual participant data, the study protocol, statistical analysis plan, informed consent and analytic codes.

**Contributors** All authors contributed to the design of this protocol. LB conceived the randomised controlled trial. GJC provided methodological expertise. JR completed the sample size calculation and designed the plan for statistical analysis. LB wrote the first draft of this manuscript and all authors gave critical feedback for revisions of this manuscript and approved the final version of this manuscript.

**Funding** This work is supported by NHS Education Scotland AHP Careers Fellowship scheme (Ref E42002/ENES7705) and NHS Research Scotland Career Researcher Fellowship scheme (Ref NRS/23/NM03).

**Competing interests** None declared.

**Patient and public involvement** Patients and/or the public were involved in the design, or conduct, or reporting, or dissemination plans of this research. Refer to the Methods and analysis section for further details.

**Patient consent for publication** Not applicable.

**Provenance and peer review** Not commissioned; externally peer reviewed.

**ORCID iDs**
Jim Richards http://orcid.org/0000-0002-4004-3115
Graham J Chapman http://orcid.org/0000-0003-3983-6641

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
