## [Reviewer comments · BMJ Open]

ARTICLE DETAILS

TITLE (PROVISIONAL)	Comparing the effectiveness of computer-aided-design/computer-aided-manufacture (CAD/CAM) insoles manufactured from foam-box cast versus direct scans on patient reported outcome measures: A protocol for a double-blinded, randomised controlled trial
AUTHORS	Barr, Laura; Richards, Jim; Chapman, Graham

VERSION 1 – REVIEW

REVIEWER	Telfer, Scott University of Washington
REVIEW RETURNED	13-Oct-2023

GENERAL COMMENTS	This manuscript presents a protocol for an RCT of foot orthoses, comparing 2 methods of obtaining foot shape. There are a few things I'd like to see added to ensure the protocol is fully described. Page 3 line 61 - Any data on what the current split between these methods is? Page 3 line 64 - Is this common? I'm aware of centers that mail foam boxes that are then scanned but not that cast in house then mail them to the manufacturer? Page 5 line 131 - If I understand the inclusion criteria correctly, you'll include a fairly broad range of patients with different conditions? Potentially going from those getting FOs for, say, plantar fasciitis to those with significant deformities as a result of rheumatoid arthritis? In addition, are there any exclusion criteria related to history of FO wear? Page 7 line 139 – I would like to see much more information on SOP for obtaining foam box and scans needed. Foam boxes especially are notoriously variable between users so exact protocol given to the clinicians obtaining the impression need to be provided. Similarly, are the Paromed scans weightbearing, is the foot held in a neutral position etc? It sounds like there will be multiple orthotists taking the scans (?), will any standardized training take place before the study begins in order to minimize differences due to shape capture technique? And what about the CAD process, how standardized is this? Page 7 line 162 - Are there other events that would necessitate the participant being withdrawn during the trial? I.e. an unrelated foot injury? What is the protocol in this case?
--

	Page 8 line 193 - Will the patient data be used if they do not meet the adherence threshold? Is the threshold an average over the 12 weeks or if they miss it for any single week?
--	--

REVIEWER	Yurt, Yasin Eastern Mediterranean University
-----------------	---

REVIEW RETURNED	10-Nov-2023
-------------

GENERAL COMMENTS	The aim of the paper is well established and the text of the paper is well written. I have some concerns which are mentioned below.  • The diagnosis of your subjects is not clear. You said medical condition or lower limb biomechanical deficit, however, this range is so wide and includes many different problems. This will affect your group homogeneity. Therefore, the patients' actual diagnosis should be considered while interpreting the effect of insoles and intergroup comparisons. • Inclusion criteria: Please write what you mean with suitable footwear. • Exclusion criteria: “.....unlikely to benefit from CAD/CAM insoles.” This criteria is not clear and seems so subjective. • What is the Shore value of the EVA used in insoles? Please write. • The information about insole manufacturing is limited. You did not mention the design and manufacturing method of the insoles. • the number of references is too much. I think there are some unnecessary references. For ex. you gave 4 ref. for OPUS, please remove the extra references.
--

VERSION 1 – AUTHOR RESPONSE

Reviewer: 1

Dr. Scott Telfer, University of Washington

Comments to the Author:

This manuscript presents a protocol for an RCT of foot orthoses, comparing 2 methods of obtaining foot shape. There are a few things I'd like to see added to ensure the protocol is fully described.

Response: We thank the reviewer for the positive comments about the protocol.

Page 3 line 61 - Any data on what the current split between these methods is?

Response: This is an interesting point and to our knowledge, no research has reported this data. However, we have just conducted a cross-section survey obtained from Freedom of Information examining the number of CAD/CAM insoles provided annually across UK Orthotics services in 2021-2022, and the methods of shape-capture used in their manufacture. The data has not been analysed as yet but we aim to publish this paper in the near future.

Page 3 line 64 - Is this common? I'm aware of centers that mail foam boxes that are then scanned but not that cast in house then mail them to the manufacturer?

Response: We have amended this line to provide greater clarity to the reader (see lines 69-71), which now reads:

“Yet the continued interim stages of foam-box casting and physical transportation of these foam-box casts to manufacturers, which is required by services who do not own or have access to scanning equipment, sacrifices these benefits.”

Page 5 line 131 - If I understand the inclusion criteria correctly, you'll include a fairly broad range of patients with different conditions? Potentially going from those getting FOs for, say, plantar fasciitis to those with significant deformities as a result of rheumatoid arthritis? In addition, are there any exclusion criteria related to history of FO wear?

Response: The reviewer is correct. We chose to take a pragmatic approach with our participant recruitment and their pathological condition(s) which reflects current day-to-day clinical NHS practice. An amendment has been made to clarify this (see lines 141-144), which now reads: "In order to provide a realistic representation of day-to-day clinical practice, participants' pathology will not be limited to one specific pathology, a similar approach taken in other studies investigating orthoses for non-specific lower limb MSK pathologies [10, 18, 19]."

Response: In relation to exclusions regarding historical insole use – Participants may be considered for recruitment only if they require a new assessment for insoles (see page 7, Table 1, inclusion criteria, point 2 which reads "Referred to the NHS GGC Orthotic service requiring a new assessment for insoles"). To clarify this point, in NHSGGC this would only include participants who do not have a historical insole prescription that is appropriate for their current needs. Participants may have a historical insole prescription which no longer meets their needs, hence a new assessment is required. This would exclude patients who have a historical insole prescription which continues to meet their current needs.

Page 7 line 139 – I would like to see much more information on SOP for obtaining foam box and scans needed. Foam boxes especially are notoriously variable between users so exact protocol given to the clinicians obtaining the impression need to be provided. Similarly, are the Paromed scans weightbearing, is the foot held in a neutral position etc? It sounds like there will be multiple orthotists taking the scans (?), will any standardized training take place before the study begins in order to minimize differences due to shape capture technique? And what about the CAD process, how standardized is this?

Response: We have amended this section (see lines 163-184), which now reads: "In order to minimise any differences between casting and scanning methods, all foam-box casts and direct-digital scans for all participants will be taken by the PI who has over 15 years' experience in the assessment, shape capture and design of insoles. All foam-box casts and direct-digital scans will be taken in a semi-weight bearing position, with the participant seated, and shape-capture undertaken one foot at a time, with the contralateral foot positioned on the floor. The foot will be manipulated by the clinician into the optimal position as determined by the participant's clinical assessment, the presenting MSK pathology and the FPI, before being placed into the foam-box and the scanner. For example, in instances of pathologies affecting the medial aspect of the foot, ankle or leg, and where FPI values are between 0 and +12, an external rotational force will be applied to the participants leg by the clinician, effectively supinating the foot in the foam-box cast and on the direct-digital scanner. Conversely, in instances of pathology affecting the lateral aspect of the foot, ankle or leg, and where FPI values are between 0 and -12, an internal rotational force will be applied to the participants leg by the clinician, effectively pronating the foot in the foam-box cast and on the direct-digital scanner. Where the participant has mobility of the first ray and the insole prescription is to be designed to facilitate first ray plantarflexion, the clinician will manipulate the first ray into a plantarflexed position by applying a downward force to the first metatarsal head in the foam-box cast and onto the direct digital scanner. These example techniques described for positioning the leg, foot and first ray, are similar to those described in previous literature regarding the effectiveness and repeatability of casting and scanning techniques [10, 26]."

Page 7 line 162 - Are there other events that would necessitate the participant being withdrawn during the trial? I.e. an unrelated foot injury? What is the protocol in this case?

Response: We have added more detail on reasons for participant withdrawal and the protocol for managing missing data in relation to this (see lines 278 - 288) which reads:

“Withdrawal of Participants from Study

During the course of the trial a participant may choose to withdraw from the trial at any time. This may happen for a number of reasons, including but not limited to the occurrence of what the participant perceives as an intolerable adverse event, inability to comply with trial procedures, or participant decision without reason. In addition, the PI may discontinue a participant from the trial treatment at any time if the PI considers it necessary for any reason including, but not limited to ineligibility arising during the trial i.e. development of a medical condition as outlined in the exclusion criteria, or significant non-compliance with treatment regimen or trial requirements i.e. participant has not worn or unable to wear the insoles between appointments. The type of withdrawal and reason for withdrawal will be recorded in the CRF.”

Page 8 line 193 - Will the patient data be used if they do not meet the adherence threshold? Is the threshold an average over the 12 weeks or if they miss it for any single week?

Response: We have added more detail regarding the calculation and analysis of data relating to the adherence threshold, see lines 295 – 298, which now read:

“Data for participants who do not reach the minimum self-reported adherence threshold of >21 hours per week, calculated as an average across the 4-week, 8-week, and 12-week time points, will still be included in the final between-group analysis to establish if adherence differs between groups.”

Reviewer: 2

Dr. Yasin Yurt, Eastern Mediterranean University

Comments to the Author:

The aim of the paper is well established and the text of the paper is well written. I have some concerns which are mentioned below.

Response: We thank the reviewer for the positive comments about the protocol.

- The diagnosis of your subjects is not clear. You said medical condition or lower limb biomechanical deficit, however, this range is so wide and includes many different problems. This will affect your group homogeneity. Therefore, the patients' actual diagnosis should be considered while interpreting the effect of insoles and intergroup comparisons.

Response: The reviewer is correct in that our inclusion criteria will affect group homogeneity.

However, we chose to take a pragmatic approach with our participant recruitment and their pathological condition(s) which reflects current day-to-day clinical NHS practice (see lines 141-144), which reads:

“In order to provide a realistic representation of day-to-day clinical practice, participants' pathology will not be limited to one specific pathology, a similar approach taken in other studies investigating orthoses for non-specific lower limb MSK pathologies [10, 18, 19].”

- Inclusion criteria: Please write what you mean with suitable footwear.

Response: We have now added 'outdoor footwear' to this inclusion criteria. In essence, the footwear needs to be able to accommodate the foot and the insole without causing any additional pain. See page 7, Table 1, Inclusion criteria, point 5, which now reads:

“Have suitable own outdoor footwear that can accommodate a CAD/CAM insole as assessed by the PI or Co-I, and can wear these for 12-weeks in accordance with standard practice”

- Exclusion criteria: “.....unlikely to benefit from CAD/CAM insoles.” This criteria is not clear and seems so subjective.

Response: We have updated this exclusion criteria referencing the NHS pathway which provides greater clarity (see exclusion criteria, page 7, Table 1, point 8), which now reads:

“Clinical assessment concludes that the participant does not require or will be unlikely to benefit from CAD/CAM insoles, as outlined in the NHS GGC Foot and Ankle Pathway [17].”

- What is the Shore value of the EVA used in insoles? Please write.

Response: We have included a description of the shore density for the insoles (see lines 218-224), which now reads:

“The EVA Shore hardness will be determined by the individual characteristics of the participant assessment. Participants with moderate to high supination resistance score or medially deviated subtalar joint axes will be considered for harder EVA insoles (50 – 70 Shore). Those participants with a low suination resistance score will be considered for 30 – 40 Shore. Participants with characteristics such as forefoot plantar fat pad atrophy will be considered for mixed 30/50 or 50/70 Shore EVA where the Shore harness at the forefoot is softer.”

- The information about insole manufacturing is limited. You did not mention the design and manufacturing method of the insoles.

Response: we have amended the broad processes for insole design (see lines 201-218), which reads:

“All scanned CAD/CAM images will then be modelled by the PI, who is blinded to the treatment arm, and has over 15 years’ experience using the Paro360 CAD/CAM modelling system. The insole prescription and design will be conferred by the PI and the Co-I who assessed the participant. The authors acknowledge that it is not possible to design a prescription protocol due to the variety of presentations that will be recruited to the trial. In accordance with standard practice, insole prescription will be determined by the physical and biomechanical assessment for each participant, and will be conferred by two experienced clinicians at the time of assessment. Prescriptions are likely to include a variety of functional design features, for example this may include the use of medial heel wedging for participants presenting with medial foot, ankle or lower limb pathology [27, 28], and medial heel skives may be considered if participants do not present with plantar heel pain [29]. Medial forefoot wedges will be considered for participants presenting with medial foot or ankle pathology and a concurrent forefoot varus [30]. Conversely the use of lateral forefoot wedges may be included for participants with a lateral foot or ankle pathology [31]. Heel raises will be considered where there is reduced range of ankle dorsiflexion, posterior or plantar heel pain, or leg length discrepancy [32]. Metatarsal domes may be considered in conjunction with other functional design elements for participants with plantar forefoot pathology [33].”

- the number of references is too much. I think there are some unnecessary references. For ex. you gave 4 ref. for OPUS, please remove the extra references.

Response: We have now reduced the number of references

VERSION 2 – REVIEW

REVIEWER	Telfer, Scott University of Washington
REVIEW RETURNED	16-Jan-2024

GENERAL COMMENTS	The authors have addressed my concerns. Best of luck with the study!
--

REVIEWER	Yurt, Yasin Eastern Mediterranean University
REVIEW RETURNED	31-Jan-2024

GENERAL COMMENTS	I thank the Authors for the corrections. I would like to recommend once again to consider homogeneity in comparisons between groups. Stratifying subjects according to their medical problems could be a way to create groups with similar problems.
---